# Identification and Analysis of the Catalase Gene Family Response to Abiotic Stress in *Nicotiana tabacum* L.

**Zhonghui Liu †, Di Wang †, Heng Tang, Haozhen Li, Xiaohua Zhang, Shaolin Dong, Li Zhang and Long Yang ***

Agricultural Big-Data Research Center, College of Plant Protection, Shandong Agricultural University, Taian 271002, China
* Correspondence: lyang@sdau.edu.cn
† These authors contributed equally to this work.

**Abstract:** Catalase (CAT) is an enzyme encoded by the catalase gene family that plays an important role in the removal of reactive oxygen species. In this study, seven CAT genes were identified in *Nicotiana tabacum* L. and were classified into three groups. Gene structure analysis revealed that *NtCAT1–6* has six or seven introns while *NtCAT7* only contains one. The relative position of introns in *NtCAT1* and *NtCAT2* had high similarity. Tissue-specific analysis shows that *NtCAT1–4* were expressed intensively in the shoot while *NtCAT5* and *NtCAT6* were in the root. *NtCAT7* expression was influenced by circadian rhythms. *NtCATs* expression had the greatest change under drought stress. Additionally, expression of *NtCAT5*, *NtCAT6* and *NtCAT7* were upregulated under cold stress but downregulated under drought and salt stress. This study will help in understanding the behavior of CAT genes during environmental stress in tobacco.

**Keywords:** *Nicotiana tabacum* L.; catalase; gene family; environmental; stress

## 1. Introduction

Various abiotic stress conditions such as cold, drought, flooding and high temperature are potential sources of ROS in organisms. Organisms also form highly active and reduced oxygen metabolites such as superoxide anions and $H_2O_2$ during cellular respiration. ROS play a dual role in plants. At low concentrations, $H_2O_2$ acts as a signaling molecule. When ROS are overgenerated in cells, they will participate in various harmful processes in organisms, such as destroying cell structures including lipids, membranes, proteins and DNA. In order to regulate oxidative stress and keep ROS at a basic nontoxic level, eukaryotic cells produce different ROS-scavenging enzymes, such as superoxide dismutase, glutathione peroxidase, and catalase. These three enzymes play an important role in preventing the key defense mechanisms of DNA, protein and lipid oxidative modification by blocking the chain reaction of free radicals in organisms. Therefore, the resistance of plants to environmental stress is related to their antioxidant systems. Genome information and methods greatly promote the understanding of plant antioxidant systems. Compared with other peroxidases, CAT has a unique advantage in removing $H_2O_2$, and its activity could help cells resist oxidative stress [1]. CAT can be used as an indicator of plant antioxidant capacity.

CATs, together with SOD and POD, are called enzyme protection systems. Their main function is to decompose hydrogen peroxide into water and oxygen, thereby removing $H_2O_2$ produced during plant photorespiration, fatty acid β-oxidation, and mitochondrial electron transport. Previous studies have shown that enhancing the activity of antioxidant enzymes and the level of antioxidant metabolism in plants can improve the stress resistance of the plants themselves. Typical CATs are homotetrameric haem proteins. They regulate hydrogen peroxide ($H_2O_2$) homeostasis by converting $H_2O_2$ into $H_2O$ and $O_2$ with high specificity for $H_2O_2$. $H_2O_2$ has toxicity because it can produce highly reactive hydroxyl radicals. CAT needs to remove excess $H_2O_2$ to maintain intracellular stability [2]. Unlike other

peroxide-metabolizing enzymes, CAT catalyzes the disproportionation reaction without a reductant but with weak activity against organic peroxides [3].

The CAT gene family has been identified in some species and expressed in different tissues and organs in plants. CAT is encoded by a small multigene family, such as seven members in cotton [4], four in cucumber [5], three in *Arabidopsis thaliana* [6] and pumpkin [7], and one in Scots pine [8]. According to the structure and functions of CAT genes, they can be divided into three groups related to photosynthetic, vascular, and reproductive functions [6,9–11]. In previous studies, *NpCat1*, *NpCat2* and *NpCat3* belonged to class I, class II, and class III, respectively; *AtCAT2* was class I [12]. According to the distance between *Nicotiana tabacum* and *N. plumbaginifolia*, the classification of tobacco CAT genes was determined [13]. In *A. thaliana*, *CAT1* and *CAT3* mRNA was most abundant in bolts and *CAT2* was highly abundant in leaves [6]. In rice seedlings, *CatA* and *CatC* were mainly expressed in leaves while *CatB* was expressed in roots [14]. In maize, the CAT-1 catalase isozyme was expressed in milky endosperm, aleurone and scutellum [15]. In hot peppers, *CaCat1* showed stronger expression in vascular tissues, while *CaCat2* was nearly equal in all tissues. In contrast to *CaCat1* and *CaCat2*, *CaCat3* demonstrated a constant low level in young seedlings and vegetative organs [16]. Three kinds of expressed CAT genes were found in *N. plumbaginifolia*, and *Cat2* mRNA levels rose rapidly when exposed to UV-B light, ozone and sulfur dioxide ($SO_2$) [17].

CAT gene expression in plants is regulated temporally. In *A. thaliana* seeding, *CAT3* and *CAT2* are controlled by the circadian clock and are mainly expressed in the evening and early morning, respectively [18]. *CAT2* and *CAT3* could maintain reactive oxygen species (ROS) homeostasis in light and dark, respectively. In rice, the peak of *CatC* mRNA occurred in the light period and earlier than *CatA*, while *CatB* occurred in the dark period. Further research suggests that *CatA* and *CatC* have different transcriptional regulatory mechanisms; one is dependent on the circadian clock while the other has a light-dependent mechanism [19].

CAT genes play an important role in plants' resistance to adversity. In *A. thaliana*, CAT plays an important role in programmed cell death and stomatal regulation under drought stress, and *CAT2* can be activated under cold and drought. The activity of *CAT3* can be enhanced by Abscisic acid (ABA), particularly by an oxidation reaction at the senescence stage [20–23]. In rice, *CatB* was significantly expressed under drought conditions [19]. When CAT of *Escherichia coli* was introduced into tobacco chloroplasts, the plants could tolerate high irradiation under drought conditions, while the photosynthesis of wild plants was seriously destroyed under the same conditions [24]. The level of CAT gene expression dropped sharply, maybe as a result of necrotic leaves and raised salicylic acid levels, and it approximated the reason for the defensive reaction [25]. In pumpkin, *cat1* encoding CAT may be involved in the senescence process [7]. The expression level of CAT2 mRNA increased rapidly under stress conditions of UV-B, $O_3$ and $SO_2$ [13].

Abiotic stresses such as cold, drought, salt and heavy metals have a great impact on the growth and development of tobacco and the quality of tobacco leaves. Exploring the mechanism of gene family response to stress in tobacco and its role in tobacco growth and development is helpful for tobacco resistance breeding. At present, the CAT gene family has been identified in many species, but there is a lack of research in tobacco. The increase in genomic data release and sequencing types in *N. tabacum* has provided a lot of information for research on the CAT gene family. In this study, CAT genes were systematically identified in *N. tabacum*, and the physiological and biochemical characteristics of each gene, gene structure, conserved domains, and phylogenetic relationships with other species were developed. In addition, the expression of CAT genes was studied in different tissues and under different abiotic stresses. The results were helpful in understanding the role of CAT in tobacco growth and development.

## 2. Materials and Methods

### 2.1. Plant Materials and Data Sources

Tobacco plants (*N. tabacum* cv. K326) were grown hydroponically in a growth chamber at 25 °C with an illumination of 16/8 h light/dark. For polyethylene glycol (PEG) and NaCl treatments, 7-week-old seedlings were transferred to growth media supplemented with 15% PEG6000 and 200 mM NaCl [26]. Cold treatments were performed by transferring 7-week-old seedlings to a pre-cooled medium in a growth chamber at 4 °C. The second and third leaves on the top were collected 24 h after each treatment. Harvested leaves were immediately frozen in liquid nitrogen and then stored at −80 °C for RNA isolation.

Total RNA of *N. tabacum* cv. K326 were isolated using a Plant Total RNA Extraction Kit (BioFlux, Beijing, China). RNA samples were reverse transcribed according to the instructions of ReverTra Ace qPCR RT Master Mix with gDNA Remover (TOYOBO, Osaka, Japan).

The K326 genome and *A. thaliana* genome were downloaded from the Solanaceae Genomics Network (SGN) (https://solgenomics.net/) (accessed on 16 April 2022) [27] and Arabidopsis Information Resource (TAIR) (https://www.arabidopsis.org/) (accessed on 16 April 2022) [28], respectively. The gene expression profile files (SRP101432) of 8-week-old tobacco roots, stems and branch tips at different time points (0 h, 6 h, 12 h and 18 h) can be obtained from NCBI [29].The protein sequences encoded by catalase genes of *Solanum lycopersicum*, *Solanum tuberosum*, *Capsicum annuum* and *Nicotiana plumbaginifolia* were also downloaded from NCBI.

### 2.2. Identification of CAT Members in Tobacco

Acquisition protein sequences of tobacco CAT genes were used as a query for BLAST with protein sequences of *A. thaliana* CAT genes, and the expected value was set to $1 \times 10^{-5}$. Then, the Hidden Markov Model (HMM) of the CAT (PF00199) obtained from the Pfam database (https://pfam.xfam.org/) (accessed on 20 April 2022) [30] was scanned with the protein sequence file to select tobacco CAT genes. Additionally, the results of BLASTP and HMM were merged and the redundant protein sequences were removed. Finally, the identified candidate CAT genes were confirmed by CD-Search in NCBI (https://www.ncbi.nlm.nih.gov/) (accessed on 26 April 2022) [31] and InterProScan (https://www.ebi.ac.uk/interpro/) (accessed on 26 April 2022) [32].

### 2.3. Phylogenetic Analysis

MEGA X v10.2.6 [33] was used to construct an unrooted phylogenetic tree using the neighbor-joining method. An NJ tree was constructed through ClustalW with 1000 bootstrap replicates and 95% partial deletion. After the phylogenetic tree was completed, the final visualization was done with Evolview (http://www.evolgenius.info/evolview) (accessed on 1 June 2022).

### 2.4. Gene Structure Analysis

CAT gene structures were analyzed using the Gene Structure Display Server (GSDS) (http://gsds.cbi.pku.edu.cn/) (accessed on 20 June 2022) [34]. The structures of genes shown with the data from gene annotation files were extracted according to the corresponding requirements.

### 2.5. Conserved Motif Prediction and Domain Presentation

The proteins of CAT were searched for conserved motifs using MEME (http://meme-suite.org/) (accessed on 21 June 2022) [35]. The motif site selected any number of repetitions (anr). The conserved domains of the protein sequences that had been identified were presented by IBS (http://ibs.biocuckoo.org/) (accessed on 21 June 2022) [36].

### 2.6. Expression Profiling Analysis by Quantitative Real-Time PCR

The transcriptome data of tobacco CAT genes were used to generate heat maps with R language. Quantitative real-time PCR (qRT-PCR) was performed to analyze the expression patterns of tobacco CAT genes. β-actin was used as an internal control. The PCR primers (Supplementary Table S1) were designed to avoid the conserved region and to amplify products of 80 to 200 bp. The qRT-PCR was performed in optical 96-well plates (Monad, Suzhou, China) with a CFX96 Touch Real-Time PCR Detection System (BIO-RAD, Hercules, CA, USA) by using 2× Universal SYBR Green Fast qPCR Mix (ABclonal, Wuhan, China) and using the following thermal cycles: 95 °C for 3 min, followed by 40 cycles at 95 °C for 5 s, and 58 °C for 30 s. Quantitative analysis was performed using the $2^{-\Delta\Delta CT}$ method [37]. Three biological duplicates were included in each treatment and there were three technical replicates for each of the biological duplicates.

### 2.7. Gene Ontology (GO) and KEGG Orthology (KO) Enrichment Analysis

The Tobacco GO annotation file was downloaded from agriGO (http://systemsbiology. cau.edu.cn/agriGOv2/download.php) (accessed on 10 July 2022) [38]. The KO term was obtained from the KAAS (KEGG Automatic Annotation Server) database (https://www. genome.jp/tools/kaas/) (accessed on 12 July 2022) [39] and the method for assigning orthologs was BBH (bi-directional best hit) with prepared protein sequences.

### 2.8. Secondary and Tertiary Structures of CAT Proteins

The secondary structures were predicted by Phyre2 programs (http://www.sbg.bio.ic. ac.uk/phyre2/html/page.cgi?id=index) (accessed on 20 July 2022). The modelling mode was normal and the PDB file could be obtained. The tertiary structures were predicted by PDBsum (http://www.ebi.ac.uk/thornton-srv/databases/pdbsum/Generate.html) (accessed on 21 July 2022) and displayed by PyMOL software.

## 3. Results

### 3.1. Identification of Tobacco CAT Genes

HMMER and BLASTP were used to search for CAT genes in tobacco protein with the HMM profile of CAT (PF00199) obtained from the Pfam database. Then, seven CAT genes were identified by using CD-Search in NCBI and named based on their proximity to *A. thaliana* (Table 1).

**Table 1.** The members of CAT gene family in *Nicotiana tabacum* L.

| Gene Name | Gene ID | Length (aa) | MW (kDa) | pI | GRAVY | Subcellular Localization |
|---|---|---|---|---|---|---|
| *NtCAT1* | Nitab4.5_0000702g0130.1 | 492 | 56.824 | 6.70 | −0.568 | pero, chlo, cyto |
| *NtCAT2* | Nitab4.5_0006960g0010.1 | 492 | 56.822 | 6.86 | −0.564 | pero, chlo, cyto |
| *NtCAT3* | Nitab4.5_0009821g0010.1 | 476 | 55.254 | 6.50 | −0.531 | cyto, pero, plas, cysk_plas, nucl |
| *NtCAT4* | Nitab4.5_0000407g0130.1 | 492 | 56.824 | 6.60 | −0.518 | cyto, pero, mito, plas, cysk_plas |
| *NtCAT5* | Nitab4.5_0000588g0030.1 | 492 | 56.947 | 6.95 | −0.569 | pero, mito, chlo, cyto |
| *NtCAT6* | Nitab4.5_0011405g0020.1 | 490 | 56.673 | 6.75 | −0.555 | pero, mito, chlo, cyto |
| *NtCAT7* | Nitab4.5_0009395g0030.1 | 91 | 10.249 | 9.25 | 0.349 | plas, golg_plas, golg |

aa: animo acid; kDa: 1000 Daltons; pero: peroxisome; chlo: chloroplast; cyto: cytoplasmic; plas: plasma membrane; cysk: cytoskeleton; nucl: nuclear; mito: mitochondrial; golg: Golgi apparatus.

CAT genes in tobacco encoded 91 to 492 amino acids and the molecular weight fluctuation range was from 55.254 kDa to 56.947 kDa, and different *NtCATs* had no different variants except *NtCAT7*, which was 10.249 kDa. The isoelectric point (pI) of CATs trended acidic. The grand average of hydropathy (GRAVY) showed that most of the CAT genes

were hydrophilic proteins. Subcellular localization showed that different genes encode proteins in different compartments but were mainly focused in the peroxisome.

### 3.2. Phylogenetic Analysis of NtCATs

An unrooted phylogenetic tree was constructed using MEGA X software with CAT genes from *N. tabacum* and other species (Figure 1). According to the phylogenetic tree, twenty-two CAT genes could be divided into three classes (class I, class II and class III). All of the identified *NtCATs* were named based on the N. plumbaginifolia CAT family. *NtCAT1* and *NtCAT2* belonged to class I, *NtCAT3* and *NtCAT4* belonged to class II, *NtCAT5*, *NtCAT6* and *NtCAT7* belonged to class III. Obviously, *NtCATs* had a high similarity in each group.

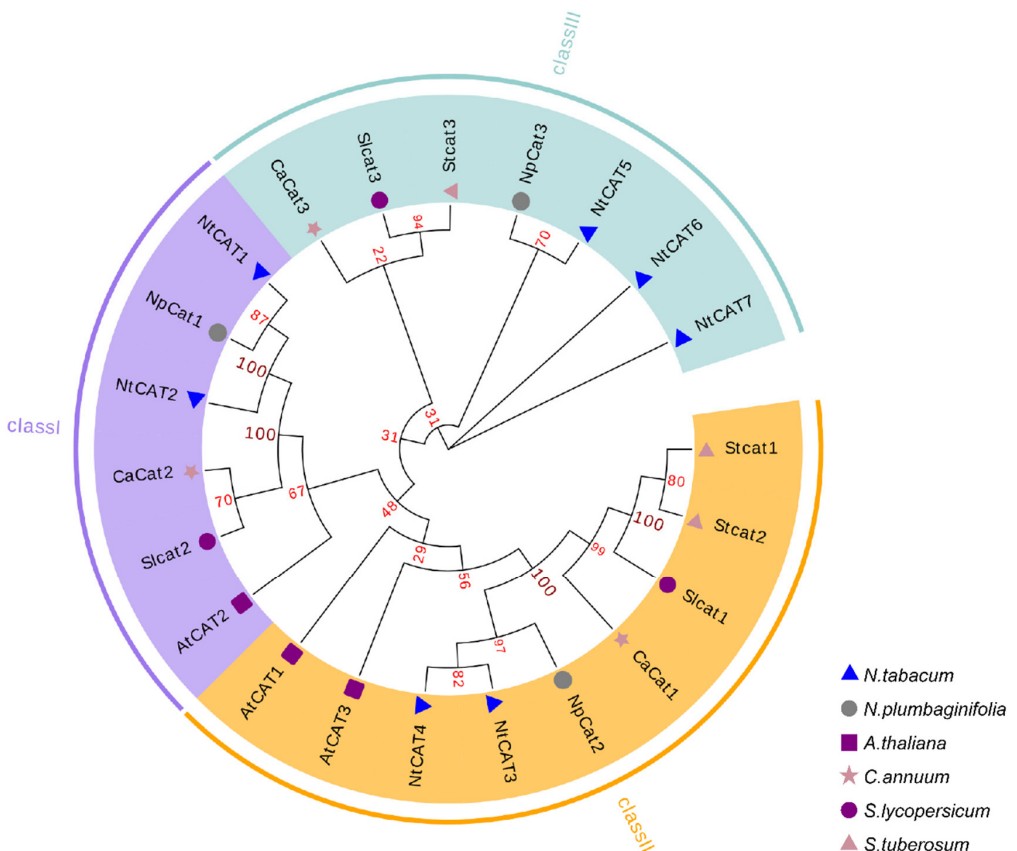

**Figure 1.** Phylogenetic relationships of CATs in six plants. GenBank accession numbers of CAT sequences were as follows: *NpCat1* (CAA85424.1), *NpCat2* (CAA85425.1), *NpCat3* (CAA85426.1), *CaCat1* (AAF34718.1), *CaCat2* (AAM97541.1), *CaCat3* (AAM97542.1), *Slcat1* (AAA34145.1), *Slcat2* (AAD41256.1), *Slcat3* (XP_004238430.1), *Stcat1* (ABC01913.1), *Stcat2* (P55312.1), *Stcat3* (XP_006342134.1).

### 3.3. NtCAT Structure Analysis

The structure of *NtCATs* was displayed by GSDS. All *NtCATs* were longer than *AtCATs*. *NtCAT1–6* had six or seven introns and four different patterns according to the positions and lengths of introns and exons. CAT genes in the same class had a highly similar pattern. The number of introns and CDS regions is the same and their distribution is similar. *NtCAT6* was the longest, but the number of introns was less than for *NtCAT1–NtCAT5*. Different from other genes, *NtCAT7* only had two CDS regions and one intron (Figure 2A,B).

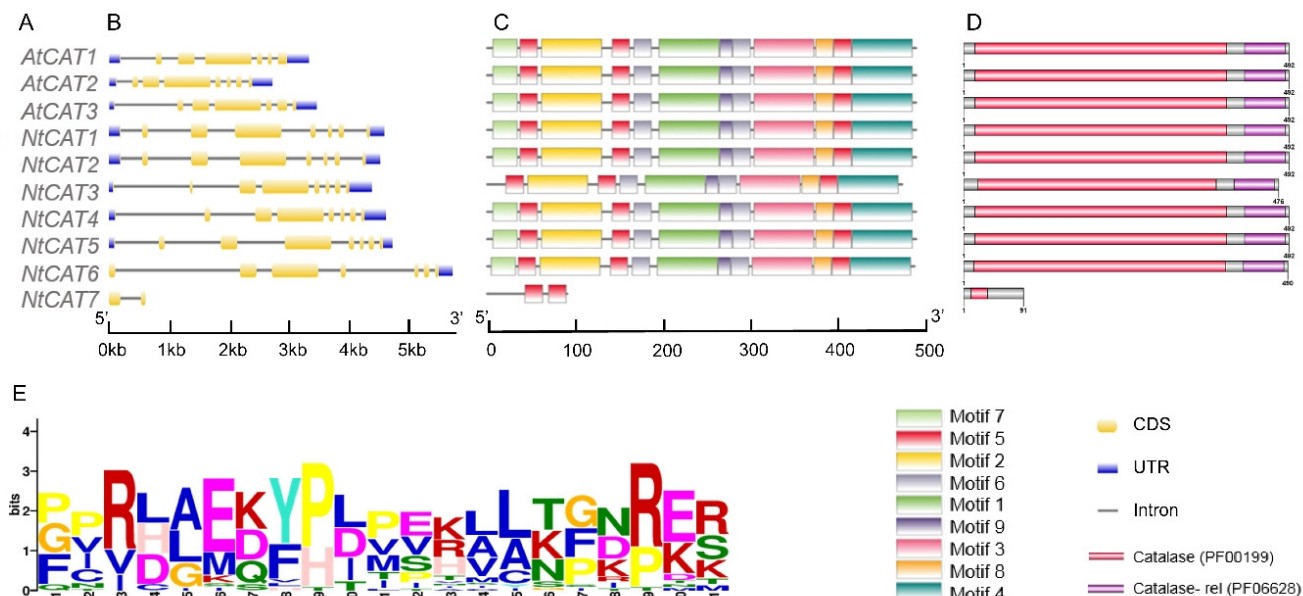

**Figure 2.** Analysis of evolutionary relationship, protein domains, multiple sequence alignment and gene structures of *NtCATs*. (**A**) Phylogenetic relationship in *AtCATs* and *NtCATs*. (**B**) The structure of *NtCATs*. The length of every region can be confirmed using the scale. (**C**) Nine conserved motifs of *NtCATs* were generated. Conserved motifs are shown in different-colored boxes. (**D**) The distribution of catalase core domain and catalase-related immune-responsive domain. (**E**) The sequences of Motif 5.

*3.4. NtCATs Conserved Motif Prediction*

Most CAT genes had nine motifs and the distribution of motifs in every protein sequence was approximately the same. Nine motifs had great similarity in CAT except for in *NtCAT7* (Figure 2C). The length of nine motifs had difference. *NtCAT3* lacked Motif 7 and was shorter than *NtCAT1–6*. Meanwhile, *NtCAT7* only had Motif 5 and lacked the CAT-rel domain (Figure 2D). The lengths of the proteins were estimated using the scale. In the CAT family, Motif 5 was shared by all members, and Motifs 1-4 were longer than the others (Figure 2E).

*3.5. NtCAT Expression in Different Tissues and Times*

In order to further study the relationship between the expression and function of the CAT gene in tobacco, the transcriptome data of an 8−week−old common tobacco genome were downloaded in NCBI, and the expression of the CAT gene in different parts (roots, shoot and shoot apex) at different times (0 h, 6 h, 12 h and 18 h) was collected. The data were standardized and plotted into a heat map with an R package (Figure 3). According to the clustering results, the CAT gene expression patterns on the same branch are the same. The results showed that *NtCAT1* and *NtCAT2* were mainly expressed in the shoot, were also expressed in the shoot apex at 0 h, and the expression level in roots was very low. The expression of *NtCAT3* and *NtCAT4* was also concentrated in the shoot, but it was only expressed at 0−12 h, and the expression level was very low at 18 h. In the shoot apex, *NtCAT3* and *NtCAT4* were not significantly expressed at all time periods, and the expression level in the root was lower than that in the shoot apex. Unlike *NtCAT1−4*, *NtCAT5* and *NtCAT6* were mainly expressed in roots. *NtCAT7* is expressed in roots, the shoot and the shoot apex. Compared with tissue-specific expression, the gene is more affected by time regulation. Compared with other time periods, *NtCAT7* is more likely to be expressed at 0 h. *NtCAT1* and *NtCAT2* were also affected by time regulation. In the shoot, genes were more likely to be expressed at 0 h than at 6 h, 12 h or 18 h. In the shoot, the expression of *NtCAT2* was higher at 0 h, while the expression of *NtCAT3* was higher at

12 h. The expression of *NtCAT1* was similar at 0 h and 18 h, and the expression of *NtCAT4* was not significantly different at 0 h and 12 h.

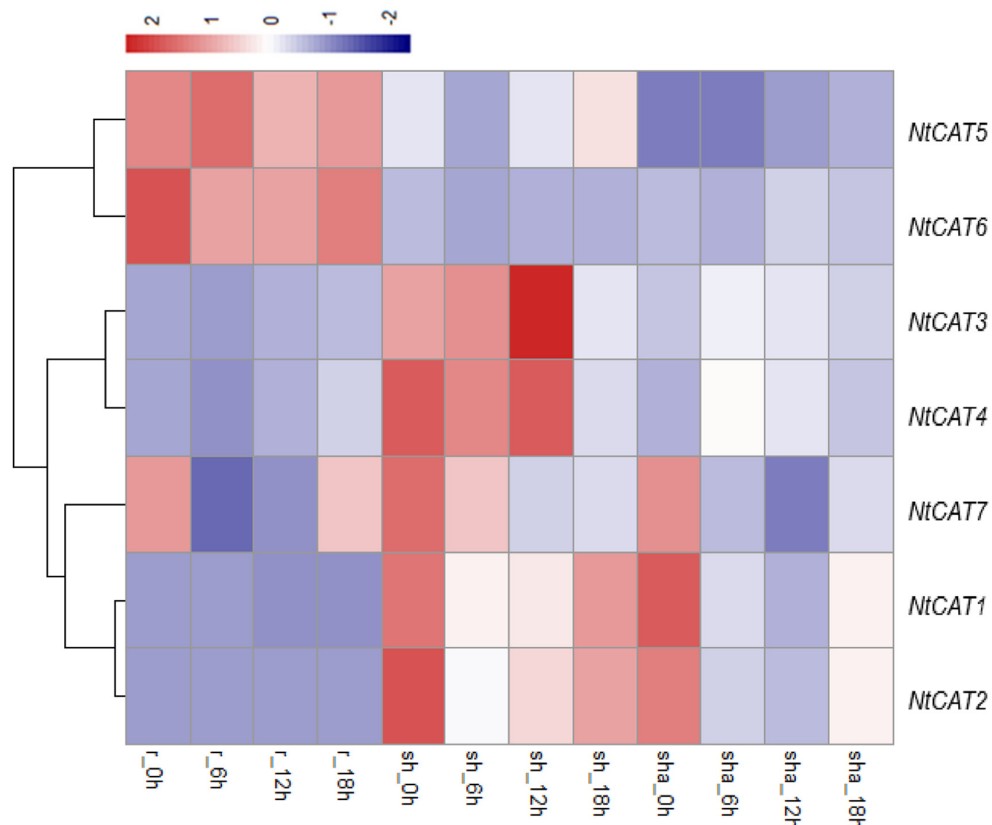

**Figure 3.** Transcriptional profiling of *NtCATs*. r: root; sh: shoot; sha: shoot apex.

### 3.6. Response of NtCATs to Various Abiotic Stresses

To investigate the response of *NtCATs* to various abiotic stresses, tobacco seedlings were treated under three different sets of abiotic stress conditions (NaCl, cold, drought) (Figure 4). Compared with the control without stress (CK), the expressions of *NtCAT1*, *2*, *3* and *4* were downregulated under abiotic stress, but were not sensitive to low temperature. The expressions of *NtCAT5*, *6* and *7* were upregulated under the stimulus of cold, while on the contrary—especially with drought and NaCl—*NtCAT6* was more sensitive to the external environment and had the greatest variation in gene expression. This suggests that gene expression was induced by cold and might have a unique regulatory pattern when confronted with cold stress. Compared with cold and NaCl stress, the gene was more sensitive to a drought environment, and the downregulation was larger, especially for *NtCAT1*, *2*, *3* and *4*; the greatest change was in *NtCAT4* expression. The experimental results showed that *NtCATs* responded to abiotic stress and were involved in the regulation of stress response.

### 3.7. NtCAT Function Analysis

GO annotation indicated that *NtCATs* were involved in molecular functions and biological processes (Figure 5). All *NtCATs* had two molecular functions, and *NtCAT7* lacked a response to oxidative stress in biological processes. Except *NtCAT7*, the others had one common pathway (K03781) and were associated with two metabolisms, two signaling pathways, one cellular process and one organismal system.

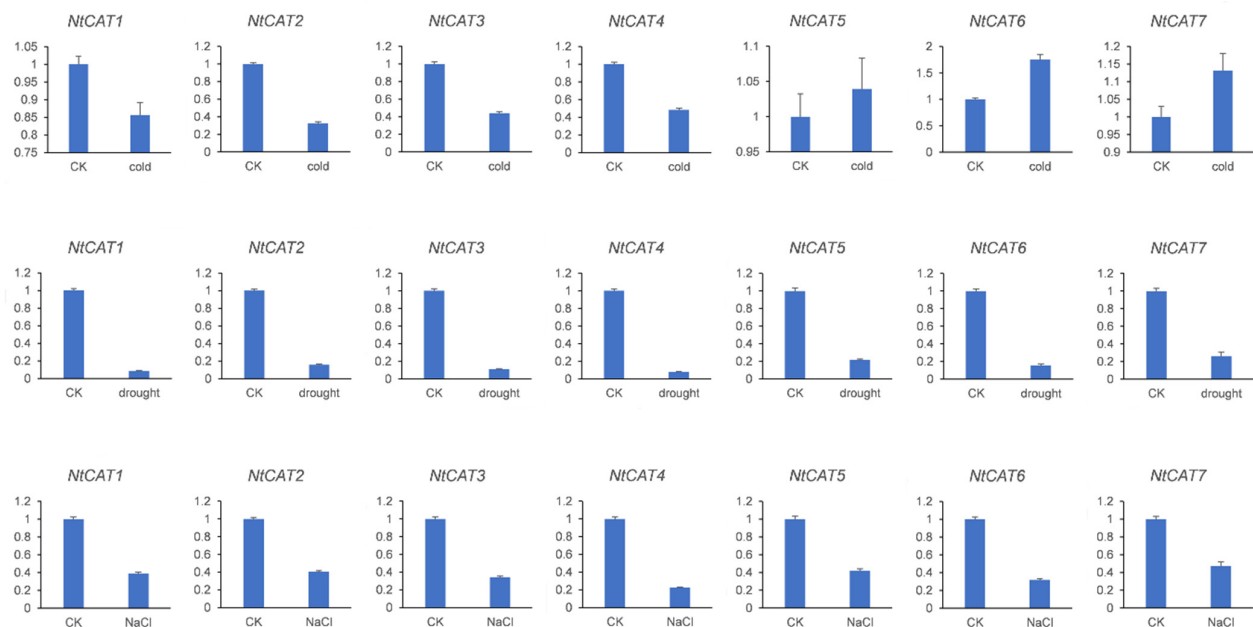

**Figure 4.** Expression patterns of *NtCATs* in response to cold, drought and NaCl treatments.

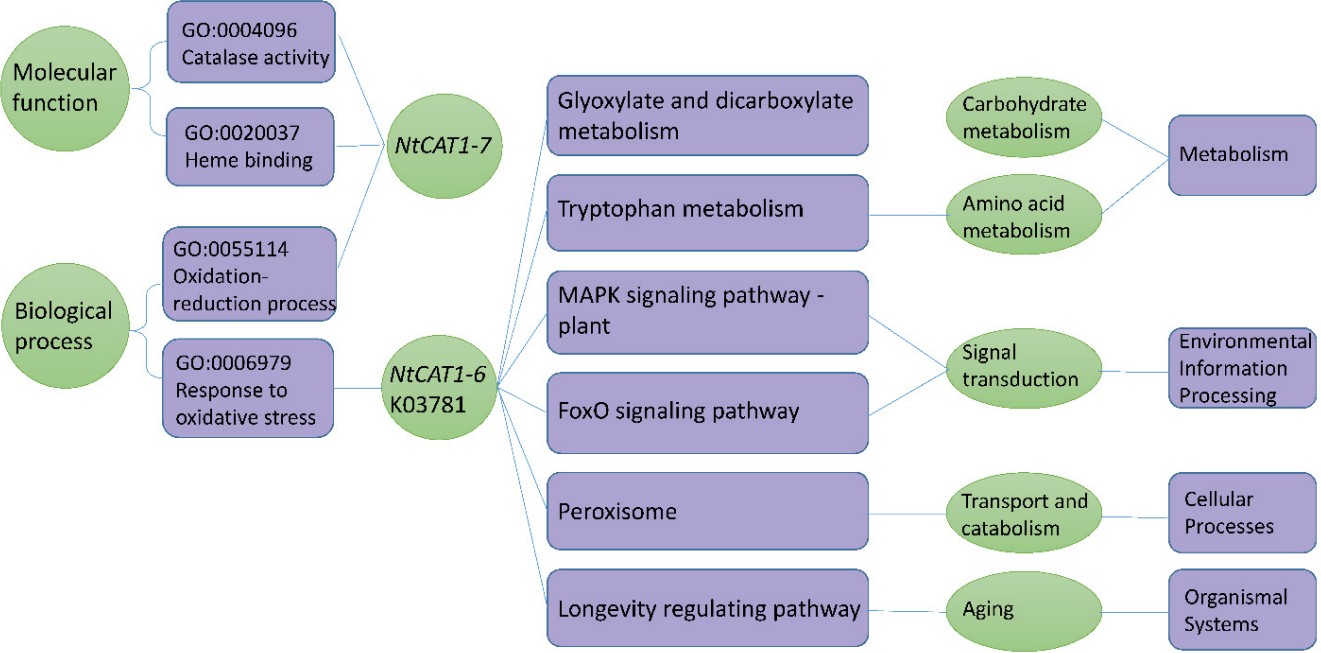

**Figure 5.** Function annotation of *NtCATs* based on Gene Ontology (GO) and KEGG pathways.

### 3.8. Secondary Structures of CAT Proteins

The secondary structure of *NtCATs* was predicted using the Phyre2 programs and these proteins revealed alpha helix, extended strand, beta turn and random coil (Figure 6). Different structures were represented by lines of different colors. The arrangement of these secondary structures in proteins was very similar. Except *NtCAT7*, random coil accounted for a large proportion (57–49%). The second was alpha helix (27–30%) and has a concentrated area in the 3′-region. Alpha helix also concentrated in the 5′-region in *NtCAT6* and *NtCAT7.* Beta turn accounted for the smallest proportion (3–6%) of secondary structures. The ratio of extended strand was higher than alpha helix in *NtCAT7*.

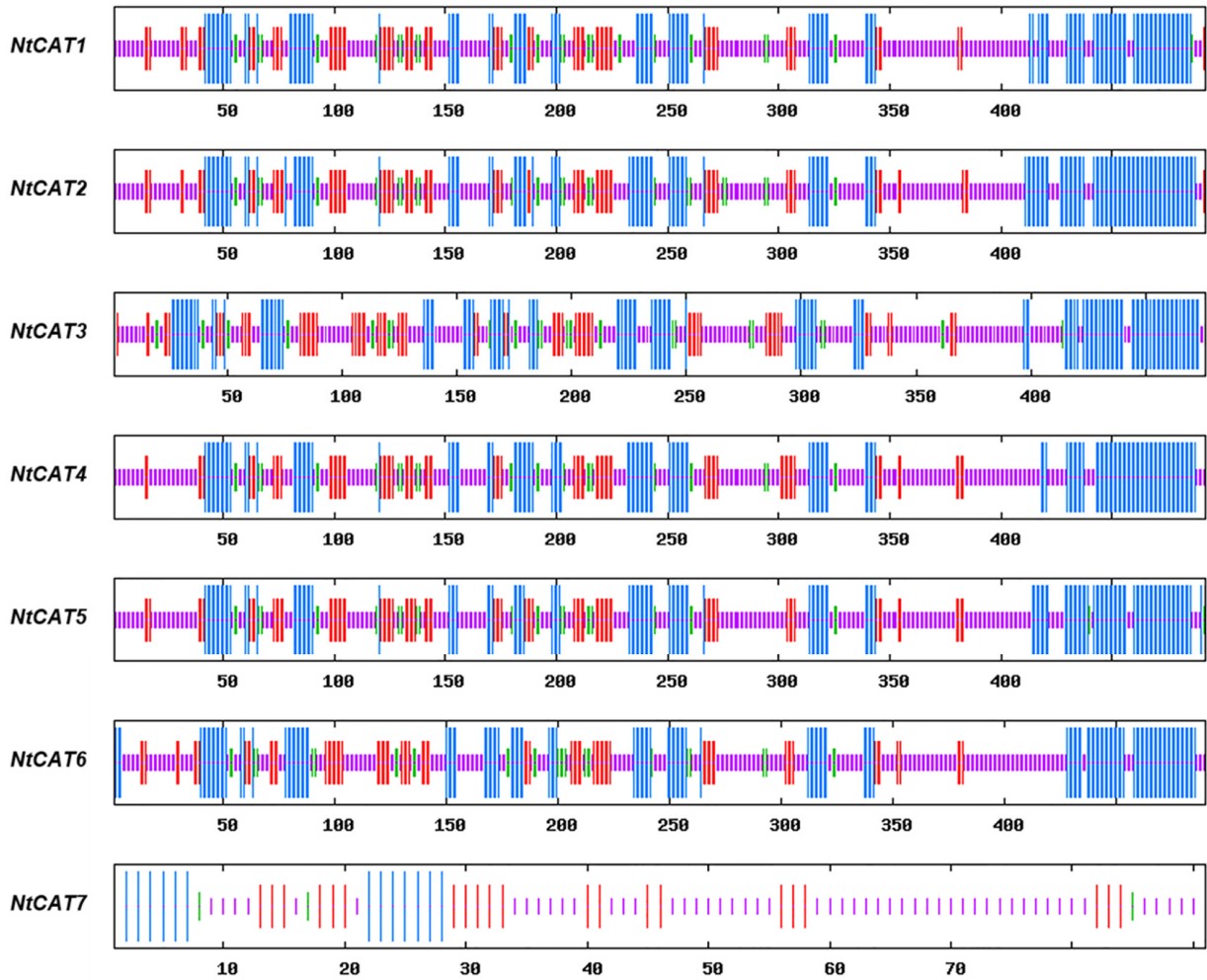

**Figure 6.** The secondary structure pattern—including alpha helix (blue color), extended strand (red color), beta turn (green color) and random coil (yellow color)—of *NtCATs*.

*3.9. Tertiary Structures of CAT Proteins*

PDBsum showed a typical three−dimensional frame comprising various parallel α−helixes. The reliability of the protein model was illustrated by ramachandran plot (Figure 7). The comparisons between these structures in *NtCATs* showed the conserved structures. The results of *NtCAT1−6* suggested that there were very few differences in structures. The structure of *NtCAT7* was different from others. Some diversities in these protein structures may reflect their different obligations in expression and function. Ramachandran plots of *NtCATs* revealed that the results of structure prediction were very reliable. Most amino acid dihedral angles appeared in the preferred region.

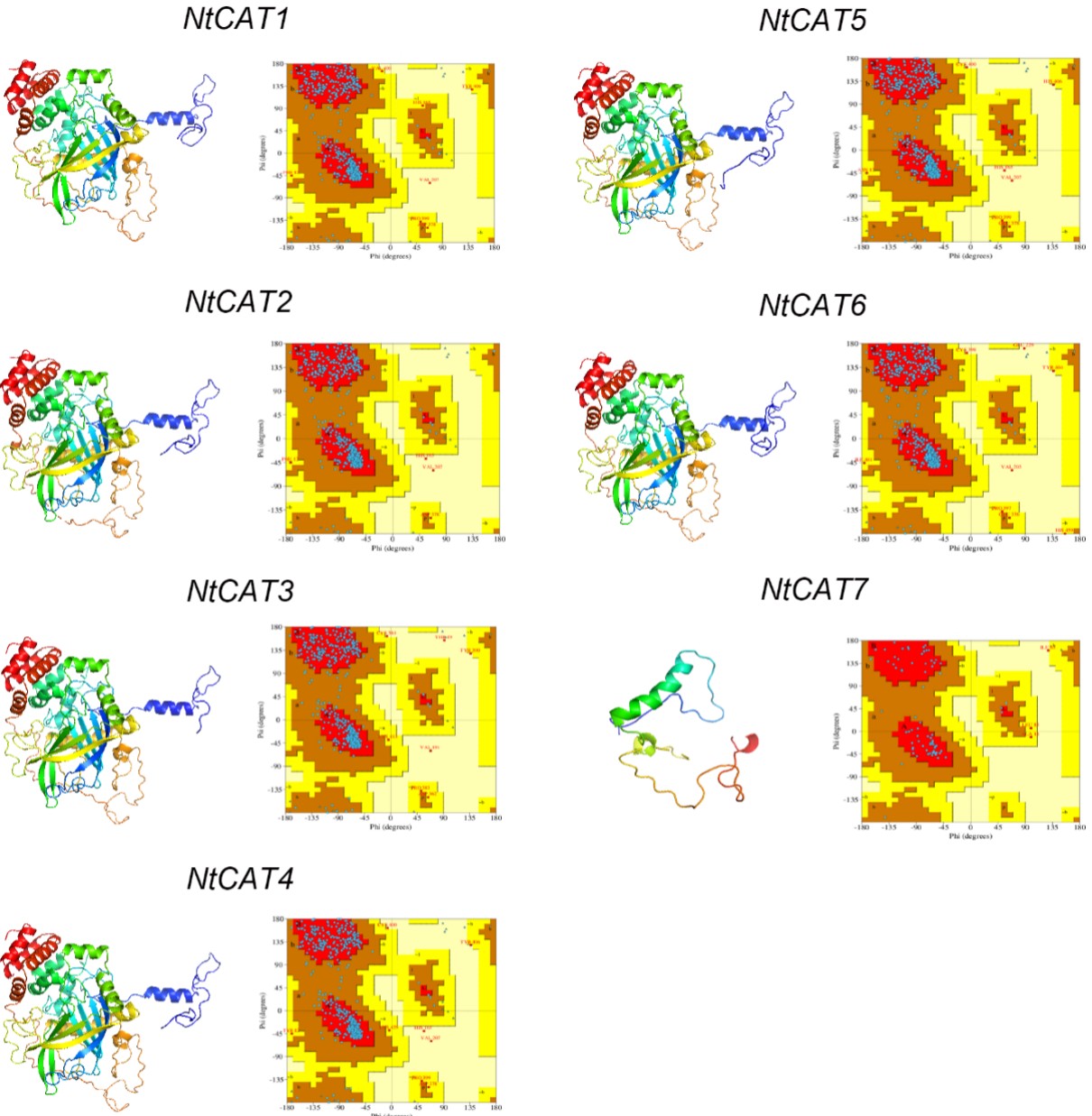

**Figure 7.** The 3D structures of *NtCAT* proteins and ramachandran plot analysis. The red, brown and yellow areas represent the preferred, permitted and generally permitted areas, respectively. The lightest areas represent the impossible.

## 4. Discussion

In this study, seven CAT genes were identified and divided into three classes (Figure 1). Only three CAT genes were found in *N. plumbaginifolia* [17]. The reason for this difference might be that *N. plumbaginifolia* was diploid, while *N. tabacum* was allotetraploid. These two types of tobacco and other solanaceae crops were used to build evolutionary trees. The phylogenetic tree analysis indicated that CAT genes in *Nicotiana* and *Solanum* are closely related. It is of great significance to study the structure of proteins. Analysis of protein structure, function and interrelationship is an important part of the proteome project. Studying the structure of proteins helps to understand the role of proteins, understand how proteins perform their biological functions, and understand the interaction between proteins. In this study, some structural diversity of CAT proteins may reflect their different responsibilities in expression and function, which lays a foundation for further analysis of

the function of the CAT gene family in tobacco (Figures 5–7). Through sequence alignment, it was found that the similarity of two protein sequences in each class is very high (>97%), except in *NtCAT7*. The results showed that CAT genes are highly conserved during evolution (Figure 2).

Three CAT genes were identified in the genomes of other species in the evolutionary tree. Variable splicing was found in three Arabidopsis family members, two kinds of splicing were found in *AtCAT2*, and four kinds of splicing were found in *AtCAT3*. A total of seven proteins were encoded by CAT genes. Variable shearing also exists in rice, with three kinds of variable shearing in *OsCATA* and two kinds of variable shearing in *OsCATB*. Studies also found that there are not only three family members in the rice genome, namely *OsCATA, OsCATB* and *OsCATC,* but also a new family member named *OsCATD.* Analysis of *OsCATD* showed that the sequence length encoded by this gene was 2392 aa, which was much longer than that of other family members. Moreover, according to the analysis of the protein domain, the protein sequence contained an AMP-binding domain (PF00501.21) that was not found in other family members [40]. This indicates that *OsCATD* gene has its own unique function while playing the role of catalase. However, no variable splicing was found in the seven family members identified, and other species' genomes may have variable splicing to consider.

CAT function analysis determined they had catalase activity and were involved in the oxidation-reduction process. Because there is more chlorophyll in shoots, *NtCAT1* and *NtCAT2* were expressed centrally in them and are related to photosynthesis. *NpCat1* was also expressed in leaves with high chlorophyll content. *NtCAT3* and *NtCAT4* expression was likely high in shoots, while *NtCAT5* and *NtCAT6* expression was high in roots (Figure 3). *NpCat3* mRNA also accumulated in flowers, and it may be related to glyoxysomal activity. In future work, the expression of *NtCAT5* and *NtCAT6* in flowers could be explored, as well as the physiological and biochemical processes involving CAT genes in different organs. Although Cat4 had been identified in *N. plumbaginifolia*, its mRNA was not detected in various organs at the time of expression analysis. The protein sequence of *NtCAT7* was very short, but it was well expressed in shoots (Figure 3). Thus, *NtCAT7* should not be considered a pseudogene.

Changes in *NtCAT* expression under cold, drought and salt stress indicate that genes respond to environmental stimuli and are involved in the defense response of tobacco (Figure 4), but CAT genes may respond differently to adversity in different species [5]. CAT genes being involved in signaling and metabolic pathways further verified that CAT genes can respond to environmental stress. The commonalities and differences in expression could be analyzed by the regulatory mechanisms of the CAT gene family in tobacco's growth and development. With the increase in availability of tobacco transcriptome data, more and more expression patterns of CAT genes may be disclosed.

## 5. Conclusions

In this study, the family members in the tobacco genome were scanned based on the identified CAT gene in the Arabidopsis genome, and the physical and chemical properties and subcellular localization of the family members were analyzed using online website tools. The phylogenetic tree of the CAT gene in tobacco and other species was constructed using MEGA X for phylogenetic analysis. Gene structure analysis, motif analysis and conserved domain analysis were performed to further understand the function of the CAT gene. Transcriptome data were used to explore the expression patterns of CAT genes under cold, drought and salt stress conditions. The conclusions are described below.

Seven gene family members were identified in tobacco and divided into three categories by phylogenetic analysis: class I, class II and class III. Gene structure analysis showed that the location and number of introns and exons of the same type of *NtCATs* were highly similar. In addition to *NtCAT7*, the remaining *NtCAT* genes contain six or seven introns. Motif analysis showed that all *NtCATs* had nine motifs except *NtCAT3* and *NtCAT7*, and Motif 5 was common to the CAT gene. Gene expression analysis showed

that the expression of different types of *NtCATs* was tissue-specific. *NtCATI–4* was mainly expressed in shoots, while *NtCAT5–6* was mainly expressed in roots. Under cold stress, the expression of *NtCAT1–4* was downregulated, while the expression of *NtCAT5–7* was upregulated. All CAT genes were downregulated under drought and NaCl stress. The changes in *NtCAT* expression under different stress conditions indicated that the genes respond to environmental stimuli and participate in the defense response of tobacco. GO and KEGG annotation of *NtCATs* showed that the genes were involved in oxidative stress response.

**Supplementary Materials:** The following supporting information can be downloaded at: https://www.mdpi.com/article/10.3390/agronomy13030936/s1, Table S1. Primers in this study.

**Author Contributions:** L.Y., Z.L. and D.W. designed the study. X.Z. and H.L. contributed to most experiments. S.D. assisted with the analysis of the results. L.Y., H.T. and L.Z. provided guidance on the study. L.Y. and L.Z. modified the language of the manuscript. Z.L., D.W. and L.Y. wrote the manuscript. All authors have read and agreed to the published version of the manuscript.

**Funding:** This work was supported by the Foundation of Shandong Province Modern Agricultural Technology System Innovation Team (SDAIT-25-01) and the Foundation of Taishan brand cigarette high-quality core raw material development and application in Shandong (202102004).

**Conflicts of Interest:** The authors declare that the research was conducted in the absence of any commercial or financial relationships that could be construed as a potential conflict of interest.

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
