# Peer review of "Identification and Analysis of the Catalase Gene Family Response to Abiotic Stress in Nicotiana tabacum L."

_agronomy, doi:10.3390/agronomy13030936_

Round 1

Reviewer 1 Report

Major revisions:

1.      The introduction part should be re-written. Now the logic of introduction is confused. Based on the current version, I cannot find why the catalase is important, why the Nicotiana tabacum should be focused, what is the basic molecular characteristics of catalase coding genes based on previously researches, etc.

2.      For discussion part, the authors should discuss their results with figures and tables.

3.      The authors should make a conclusion.

4.      The authors should discuss the biological meaning for secondary and tertiary structure analysis.

5.      For material and methods part, the authors also should carefully check it, for example, the concentration for NaHCO3 is absent in line 67.

6.      Why the classifications of catalase genes from figure 1 and figure 2A show different, the authors should carefully check them.

7.      Line 33, the reference 14 is related to hot pepper, not N. plumbaginifolia indicating the references for this manuscript is messy? The authors should check it again.

 Minor revision:

1.      Line 32, “to classI, classII, classIII successively, AtCAT2 was classI” should be “to class I, class II, class III successively, AtCAT2 was class I”, some blanks should be inserted. The authors should carefully check the all the text in this manuscript.

2.      Line 110, how many biological duplicates did the authors carried out? The authors should explain.

3.      For table 1, the full name of aa and kDa should be explain.

4.      Line 127, the ‘Nicotiana tabacum’ should be italic.

5.      Line 242, one blank should be inserted between 2392 and aa.

Author Response

Response to Reviewer 1 Comments

Point 1:Theintroduction part should be rewritten. Now the logic of introduction is confused. Based on the current version, I cannot find why the catalase is important, why the Nicotiana tabacum should be focused, what is the basic molecular characteristics of catalase coding genes based on previously researches, etc.

Response 1: All authors fully gratitude your comment and agree with what you pointing out. 

We have readjust and improved the content of the introduction part, so that the overall logic of this part is more perfect, and it can more clearly explain the purpose and significance of our study, including the importance and necessity of catalase research and why tobacco needs to be paid attention to and other details.

CAT is a specific enzyme for scavenging H2O2, which can protect cells from ROS toxicity, help plants cope with abiotic stresses such as high temperature, cold and drought, and also have a certain resistance to pathogen invasion. At present, the CAT gene family has been identified in many species, but there is a lack of research in common tobacco. In this study, the CAT gene family in common tobacco was systematically identified by bioinformatics. Exploring the expression pattern of CAT gene is helpful to understand the regulation mechanism of CAT gene under different stress conditions and improve the stress resistance of tobacco.

Abiotic stresses such as cold, drought, salt and heavy metals have a great impact on the growth and development of tobacco and the quality of tobacco leaves. Exploring the mechanism of gene family response to stress in tobacco and its role in tobacco growth and development is helpful for tobacco resistance breeding.

Thank you again for your comment sincerely

Point 2: For discussion part, the authors should discuss their results with figures and tables.

Response 2: All authors fully gratitude your comment and agree with what you pointing out. Citing pictures and charts in the discussion section can indeed make our meaning more intuitive and clear to readers. We have discussed our results by citing charts and charts in the discussion section of the latest submitted manuscript as suggested by you. Thank you again for your comment.

Comment 3:  The authors should make a conclusion.

Response 3: All authors fully gratitude your comment and agree with what you pointing out. We have added a discussion section at the end of the article, which makes our article and content more substantial and perfect. The following conclusions were drawn:

“In this study, the family members in the tobacco genome were scanned based on the identified CAT gene in the Arabidopsis genome, and the physical and chemical properties and subcellular localization of the family members were analyzed using online website tools. The phylogenetic tree of CAT gene in tobacco and other species was constructed by MEGA X for phylogenetic analysis. Gene structure analysis, motif analysis and conserved domain analysis were performed to further understand the function of CAT gene. Transcriptome data were used to explore the expression patterns of CAT genes under cold, drought and salt stress conditions. The conclusions are as follows :

   Seven gene family members were identified in tobacco and divided into three categories by phylogenetic analysis : class I, class II and class Ⅲ. Gene structure analysis showed that the location and number  of introns and exons of the same type of NtCATs were highly similar. In addition to NtCAT7, the remaining NtCATs genes contain 6 or 7 introns. Motif analysis showed that all NtCATs had 9 motifs except

NtCAT3 and NtCAT7, and motif5 was common to the CAT gene. Gene expression analysis showed that the expression of different types of NtCATs was tissue-specific. NtCATI-4 was mainly expressed in buds,

while NtCAT5-6 was mainly expressed in roots. Under cold stress, the expression of NtCAT1-4 was down-regulated, while the expression of NtCAT5-7 was up-regulated. All CAT genes were down-regulated under drought and NaCl stress. The changes of NtCATs expression under different stress conditions indicated that the genes responded to environmental stimuli and participated in the defense response of tobacco. GO and KEGG annotation of NtCATs showed that genes were involved in oxidative stress.”

 Thank you again for your comment.

Comment4:  The authors should discuss the biological meaning for secondary and tertiary structure analysis.

Response 4: All authors fully gratitude your comment and agree with what you pointing out. We have added the biological significance of studying protein secondary structure and tertiary structure in the discussion section, which can indeed enrich and complete the content of our article. We added to the manuscript as follows:

“It is of great significance to study the structure of proteins. Analysis of protein structure, function and interrelationship is an important part of the proteome project. Studying the structure of proteins helps to understand the role of proteins, understand how proteins perform their biological functions, and understand the interaction between proteins. In this study, some structural diversity of CAT proteins may reflect their different responsibilities in expression and function, which lays a foundation for further analysis of the function of CAT gene family in tobacco.”

Thanks again for your comments.

Comment 5: For material and methods part, the authors also should carefully check it, for example, the concentration for NaHCO3 is absent in line 67.

Response 5: Thank you very much for pointing this out. In the original design of the experiment, we planned to use NaHCO3 treatment as the alkali stress situation. However, when writing the manuscript, the results of the alkali stress part of the experiment had not been well presented, so we had to abandon the alkali stress treatment. Unfortunately, we forgot to delete this section when writing the Materials and methods section. This is our negligence, and now it has been deleted in the manuscript. Thank you again for pointing out this mistake.

Comment 6: Why the classifications of catalase genes from figure 1 and figure 2A show different, the authors should carefully check them.

Response 6: All authors fully gratitude your comment and attaches great importance to this issue.

Evolutionary relationships should be presented more clearly so that readers can more easily and accurately understand the results of this study. The similarity degree of CAT protein sequence is different in different species, which is the reason for the different classification of the two evolutionary trees. We have made the evolutionary tree of six species. In addition to Arabidopsis, we also include pepper, tomato and potato, which belong to the same family as common tobacco. With reference to your suggestion, we chose to keep the phylogenetic tree of only six species, so the second phylogenetic tree and related descriptions were removed from the manuscript. All authors agree that this is a better way of presentation, which can make it easier and more accurate for readers to understand the content of the article.

Thank you again for your comment.

Comment 7: Line 33, the reference 14 is related to hot pepper, not N. plumbaginifolia indicating the references for this manuscript is messy? The authors should check it again.

Response 7: Thank you for pointing out the order of the references. We corrected reference14 and reviewed all references in the manuscript. The adjusted reference citation order can be seen in our manuscript, thank you again for pointing out the errors.

Minor revision:

Comment  1Line 32, “to classI, classII, classIII successively, AtCAT2 was classI” should be “to class I, class II, class III successively, AtCAT2 was class I”, some blanks should be inserted. The authors should carefully check the all the text in this manuscript.

Comment  3For table 1, the full name of aa and kDa should be explain.

Comment  4Line 127, the ‘Nicotiana tabacum’ should be italic.
Comment  5Line 242, one blank should be inserted between 2392 and aa.

Response 1345We are very grateful to you for pointing out these format errors. We have revised the manuscript and carefully checked whether there are similar errors in the full text. Thank you again for pointing out our errors.

Comment  2Line 110, how many biological duplicates did the authors carried out? The authors should explain.

Response 2Thank you very much for pointing it out. Actually, we carried out three biological duplicates. We have added a note to this part of the manuscript and thank you again for helping us to point this out.

Reviewer 2 Report

I understand the purpose of the authors in wanting to increase knowledge regarding the diversity of genes in plants, but it does not seem to me to be an original or impactful work. Most of the results are predictions and only experimentally test the expression levels under different stress conditions. English is poor but they can improve it.

Below I write some of my specific comments

In the first paragraph of the introduction you use a lot of verbs in the past tense and I think they should be in the present tense. For example -Catalase (CAT) was-, - H2O2 had toxicity-, - CAT needed to remove-, among many others.

Line 82. Query.

The wording of lines 82 to 83 is confusing, rephrase.

Line 131. Amino

What do you use as an outgroup in your phylogenetic tree? And why do you only use six plants for the tree? You should use more, there are many plant genomes and if you put more families the grouping is probably more informative.

Figure 4. What does CK mean? I assume it's a control but I'm not sure. I said that because you said CAT1-4 are down regulated and I think CK is your control without stress. Is that correct?

Does NtCAT7 have regulatory sequences like a promoter? It appears to be just a segment of something, but it may not be functional.

Author Response

Responses to Reviewer 2

Comment  1 In the first paragraph of the introduction you use a lot of verbs in the past tense and I think they should be in the present tense. For example Catalase (CAT) was-, - H2O2 had toxicity-, - CAT needed to remove-, among many others.

Response 1All authors fully gratitude your comment and agree with what you pointing out. As you suggested, we have corrected the tense problem of the language in the manuscript, and the revised version can be viewed in the newly submitted document.

For example:

Line37 Change the words “were” file into words “are”.

Line39 Change the words “had” file into words “has”.

Thank you again for your comment.

Comment 2: The wording of lines 82 to 83 is confusing, rephrase.

Response 2:Thank you very much for pointing this out. As you suggested, we have modified the wording of the sentences in lines 82-83 for better understanding of the readers as follows:

“The gene expression profile files ( SRP101432 ) of tobacco roots, stems and branch tips at different time points ( 0h, 6h, 12h and 18h ) can be obtained from NCBI. The protein sequences encoded by catalase genes of Solanum lycopersicum, Solanum tuberosum, Capsicum annuum and Nicotiana plumbaginifolia were also downloaded from NCBI.”

Thank you again for your comment.

Comment 3

What do you use as an outgroup in your phylogenetic tree? And why do you only use  six plants for the tree? You should use more, there are many plant genomes and if you put more families the grouping is probably more informative.

Response 3All authors fully gratitude your comment. All authors believe that the phylogenetic tree of multi-species can better explain the evolutionary relationship. In the study of CAT gene family in wheat and cucumber, the phylogenetic trees of wheat and cucumber with 5 or 3 outgroup CAT genes were constructed, which revealed the evolutionary relationship between wheat and cucumber with other CAT gene families. These studies laid a foundation for further analysis of the function of the CAT gene family in wheat and cucumber [1-3].  In the process of research, a lot of information and seriously considered were consulted in the above selection. Arabidopsis thaliana, a dicot plant like tobacco, has been the most widely studied gene function as a model species. In addition, we also selected Nicotiana plumbaginifolia,  which belongs to the genus Nicotiana with common tobacco, and Solanum lycopersicum、Solanum tuberosum and Capsicum annuum, which belong to the solanaceae family with tobacco. CAT genes have been identified in all of these species, and all of these species have three members of the catalase gene family in their genomes. Evolutionary relationships and functions of the tobacco CAT gene family can be better studied based on the distribution of different branches in the phylogenetic tree and the construction of the evolutionary tree through these six species. Thanks again for your concern about the content of the manuscript.

  1. ZHANG Y, ZHENG L, YUN L, et al. Catalase (CAT) Gene Family in Wheat (Triticum aestivum L.): Evolution, Expression Pattern and Function Analysis [J]. Int J Mol Sci, 2022, 23(1).
  2. HU L, YANG Y, JIANG L, et al. The catalase gene family in cucumber: genome-wide identification and organization [J]. Genetics and Molecular Biology, 2016, 39: 408 - 15.
  3. TYAGI S, SHUMAYLA, MADHU, et al. Molecular characterization revealed the role of catalases under abiotic and arsenic stress in bread wheat (Triticum aestivum L.) [J]. J Hazard Mater, 2021, 403: 123585.

Comment 4What does CK mean? I assume it's a control but I'm not sure. I said that because you said CAT1-4 are down regulated and I think CK is your control without stress. Is that correct?

Response 4Thank you very much for pointing this out. CK here refers to the control treatment without stress, which was not stated in the original manuscript due to our negligence, we have stated in the newly submitted manuscript.

Thank you again for pointing out our mistake.

Comment 5Does NtCAT7 have regulatory sequences like a promoter? It appears to be just a segment of something, but it may not be functional.

Response 5: All authors fully gratitude your comment. Through sequence alignment, it was found that the similarity of protein sequences in the same group was very high ( > 97 % ) except for NtCAT7. This may be because the protein sequence encoded by NtCAT7 is too short, resulting in poor matching with NtCAT5 and NtCAT6. The NtCAT7 gene sequence may be incomplete, but it also contains the conserved domain of catalase, and unlike the fourth CAT gene found in Nicotiana tabacum, NtCAT7 is expressed in different tissues and organs and under different stress conditions. Therefore, we speculate that NtCAT7 does have regulatory sequences like a promoter, and can exclude the speculation that NtCAT7 is a pseudogene.

Thank you again for your comment.

Round 2

Reviewer 1 Report

1.       The conclusion part should be located behind the discussion part.

2.       As mentioned last time, the authors should discuss their results with figures and tables. For example, the sentence in line 274 should be “In this study, seven CAT genes were identified and divided them into three class (Fig. 1).”

3.       It’s hard to understand the sentences from line 102 to 105. Were the tobacco tissues treated with stresses? In this case, the figure 3 is also hard to understand. Of course, the authors can use the public data from NCBI. But they should explain the data in detail, such as how old the plants were used, how the stress treatments were carried out, etc.

4.       The reference order should be updated. Now the reference order is not match to the main text.

5.       “0h, 6h, 12h and 18h” should be “0 h, 6 h, 12 h and 18 h”. One blank should be inserted between numbers and h.

6.       Line 22 and23: “H2O2” should be “H2O2”.

7.       Line 211, “NtCAT1, 2, 3 and 4” should be “NtCAT1, 2, 3 and 4”. For gene name, the italic should be used.  

8.       Line 58-61: In my opinion, ”CAT3 expression was not light responsive but controlled by circadian clock. CAT3 and CAT2 were mainly expressed in the…” can be change to “CAT3 and CAT2 were controlled by circadian clock and mainly expressed in the …”.

Author Response

Response to Reviewer 1 Comments

Point1.  The conclusion part should be located behind the discussion part.

Response 1: All authors fully gratitude your comment and agree with what you pointing out. We have changed the order of discussion and conclusion in the manuscript, and put the conclusion behind the discussion section. Thank you again for pointing this out.

Point2.  As mentioned last time, the authors should discuss their results with figures and tables. For example, the sentence in line 274 should be “In this study, seven CAT genes were identified and divided them into three class (Fig. 1).”

Response 1: All authors fully gratitude your comment and agree with what you pointing out. We illustrate our results in the corresponding position of the discussion section by citing figures, making the contents of the discussion section more complete and easier for readers to understand. Thank you again for your comments.

Point3.  It’s hard to understand the sentences from line 102 to 105. Were the tobacco tissues treated with stresses? In this case, the figure 3 is also hard to understand. Of course, the authors can use the public data from NCBI. But they should explain the data in detail, such as how old the plants were used, how the stress treatments were carried out, etc.

Response 3: All authors are grateful for your comments. In this part we cited public data in NCBI, in the original article, the authors used the expression profile of 8-week-old tobacco seedlings at different time periods without additional stress. It is indeed our fault that we did not specify the details of the data. We have added a description for this part in the manuscript according to your suggestion, and also explained Figure 3 in more detail, hoping to make this part easier to understand. Thank you again for pointing this out.[1]

  1. EDWARDS K D, FERNANDEZ-POZO N, DRAKE-STOWE K, et al. A reference genome for Nicotiana tabacum enables map-based cloning of homeologous loci implicated in nitrogen utilization efficiency [J]. Bmc Genomics, 2017, 18(1): 448.

Point4. The reference order should be updated. Now the reference order is not match to the main text.

Response 4: All authors thank you for your comments and agree with what you have pointed out. Due to our negligence, the problem of mismatch between references and original articles was caused in the process of writing the manuscript. We again carefully checked the correspondence between the reference list and the content of the article, and revised where mistakes were made. Thank you very much for reading our manuscript carefully and giving suchvaluablecomments.Thank you again for your comments.

Point5.“0h, 6h, 12h and 18h” should be “0 h, 6 h, 12 h and 18 h”. One blank should be inserted between numbers and h.

Point6. Line 22 and23: “H2O2” should be “H2O2”.

Point7.  Line 211, “NtCAT1, 2, 3 and 4” should be “NtCAT1, 2, 3 and 4”. For gene name, the italic should be used.  

Response 5,6,7: All authors fully gratitude your comments and agree with what you pointing out. We have revised these formatting issues in the corresponding sections of the article, and thank you again for such careful review and comments.

Point8. Line 58-61: In my opinion, ”CAT3 expression was not light responsive but controlled by circadian clock. CAT3 and CAT2 were mainly expressed in the…” can be change to “CAT3 and CAT2 were controlled by circadian clock and mainly expressed in the …”.

Response 8: All authors fully gratitude your comments and agree with what you pointing out. The expression method you suggested is really more concise, without too much repetition. Based on your suggestion, we have revised the sentences in the manuscript.

Thank you again for your comments.

Reviewer 2 Report

H2O2, can you put the numbers 2 as a subscript?

I asked you for change the verbs in past, but that is not only deleted the “d”, ej. line 41 you deleted the "d" but the correct word is catalyzes.

You added new information in the introduction but some parts are redundant, the idea is to organize and integrate the whole introduction so that it is informative without being repetitive.

Author Response

Response to Reviewer 2 Comments

Point1: H2O2, can you put the numbers 2 as a subscript?

Response 1: All authors fully gratitude your comments and agree with what you pointing out. We have revised these formatting issues in the corresponding sections of the article, and thank you again for such careful review and comments.

Point2: I asked you for change the verbs in past, but that is not only deleted the “d”, ej. line 41 you deleted the "d" but the correct word is catalyzes.

Response 2: All authors fully gratitude your comments and agree with what you pointing out. We have carefully checked whether the words used in the preface are accurate, and corrected several wrong words. Thank you very much for your careful review of our manuscript, and thank you again for pointing this out.

Point3: You added new information in the introduction but some parts are redundant, the idea is to organize and integrate the whole introduction so that it is informative without being repetitive.

Response 3: All authors fully gratitude your comments and agree with your comments. Before, we added some statements in the introduction part to enrich our argument, but indeed, as you said, some information is redundant and there is a situation of tautness. According to your opinion, in order to enrich the introduction part and not repeat it, we reorganized and adjusted the information in the introduction part and unnecessary duplicate statements were removed. Thank you again for your careful review and valuable comments.
